# Immunophenotype of Measurable Residual Blast Cells as an Additional Prognostic Factor in Adults with B-Cell Acute Lymphoblastic Leukemia

**DOI:** 10.3390/diagnostics13010021

**Published:** 2022-12-21

**Authors:** Yulia Davydova, Irina Galtseva, Nikolay Kapranov, Ksenia Nikiforova, Olga Aleshina, Yulia Chabaeva, Galina Isinova, Ekaterina Kotova, Andrey Sokolov, Vera Troitskaya, Sergey Kulikov, Elena Parovichnikova

**Affiliations:** National Medical Research Center for Hematology, 125167 Moscow, Russia

**Keywords:** flow cytometry, immunophenotype, measurable residual disease, acute lymphoblastic leukemia

## Abstract

Measurable residual disease (MRD) is a well-known independent prognostic factor in acute leukemias, and multicolor flow cytometry (MFC) is widely used to detect MRD. MFC is able not only to enumerate MRD accurately but also to describe an antigen expression profile of residual blast cells. However, the relationship between MRD immunophenotype and patient survival probability has not yet been studied. We determined the prognostic impact of MRD immunophenotype in adults with B-cell acute lymphoblastic leukemia (B-ALL). In a multicenter study RALL-2016 (NCT03462095), 267 patients were enrolled from 2016 to 2022. MRD was assessed at the end of induction (day 70) in 94 patients with B-ALL by six- or 10-color flow cytometry in the bone marrow specimens. The 4 year relapse-free survival (RFS) was lower in MRD-positive B-ALL patients [37% vs. 78% (*p* < 0.0001)]. The absence of CD10, positive expression of CD38, and high expression of CD58 on MRD cells worsened the 4 year RFS [19% vs. 51% (*p* = 0.004), 0% vs. 51% (*p* < 0.0001), and 21% vs. 40% (*p* = 0.02), respectively]. The MRD immunophenotype is associated with RFS and could be an additional prognostic factor for B-ALL patients.

## 1. Introduction

Relapse of acute lymphoblastic leukemia (ALL) can be explained by the persistence of measurable residual disease (MRD) after the achievement of complete remission (CR) [1]. MRD detection is incorporated in the vast majority of clinical protocols. MRD is the strongest independent prognostic factor in both pediatric and adult ALL [2,3]. According to a meta-analysis for pediatric patients (n = 11,249), 10 year event-free survival (EFS) was better for those who were MRD-negative (77% vs. 32%). For adult patients (n = 2076), the 10 year EFS was 64% in MRD-negative cases vs. 21% for MRD-positive cases [4]. MRD is associated with a high probability of relapse and is used for risk stratification and treatment decision making such as deintensification or escalation of therapy, use of targeted therapy, or allogeneic stem-cell transplantation (ASCT) [5,6,7,8,9,10,11,12].

Multicolor flow cytometry (MFC) and polymerase chain reaction (PCR) are widely used methods for MRD detection with high sensitivity of at least 10^−4^ (0.01%) [13]. MRD monitoring is performed by real-time quantitative PCR of rearranged immunoglobulins and T-cell receptor genes in numerous studies [10,14,15,16]. This method has been highly standardized by international consortia [17]. However, this method is time- and labor-consuming, and it requires the assessment of an initial diagnostic sample. In about 5% of cases, the PCR target cannot be identified [18] or can be lost during the course of the disease [19,20].

MFC is faster, cheaper, and applicable to most ALL cases in comparison with PCR. MFC allows for the detection of MRD with two different approaches. The first is the determination of the leukemia-associated immunophenotype (LAIP) of blast cells before treatment and tracking it in follow-up samples. The second method called “different-from-normal” (DfN) requires strong knowledge of antigen expression patterns of normal hematologic progenitors, relies on differences between immunophenotype of blast cells and normal cells, and does not require a pretreatment sample [21,22]. However, MFC is less standardized than PCR. The lack of uniformity for MRD assessment between laboratories is one of the important limitations of MFC [1].

MFC allows not only to enumerate MRD accurately but also to describe the antigen expression profile of residual blast cells. However, the relationship between MRD immunophenotype and patient survival had not been studied yet. The aim of the presented study is to assess an immunophenotype significance of MRD in B-ALL patients enrolled in a Russian multicenter study.

## 2. Materials and Methods

### 2.1. Patients

From 2016 to 2022, 267 Ph-negative ALL [146 B-ALL, 109 T-ALL, and 12 mixed-phenotype acute leukemia (MPAL)] patients were included in the Russian multicenter study RALL-2016 (Russian Acute Lymphoblastic Leukemia study) (NCT03462095). It was planned to include 350 patients in the study by the end of 2022; however, due to the COVID-19 pandemic, the recruitment rate significantly decreased. Assessment of MRD as a prognostic factor was performed in patients with complete remission and in cases without significant deviations from the therapeutic protocol. Thus, MRD was evaluated in 190 ALL patients (98 B-ALL, 84 T-ALL, and eight MPAL). The median age of B-ALL patients was 33 (range: 18–55). The male-to-female ratio was 54:44. According to EGIL classification [23], 19 patients were diagnosed with B-I (pro-B), 78 were diagnosed with B-II (common) ALL, and five were diagnosed with B-III (pre-B) ALL. MRD was assessed in the bone marrow specimens at different timepoints: at the end of induction II (day 70) in 94 B-ALL patients, at the end of consolidation III (day 133)—n 78 B-ALL patients, and at the end of consolidation V (day 190)—in 76 B-ALL patients. In four patients, MRD was analyzed at 133 or 190 days but not at day 70 because of issues with sample delivery. All cases were Ph-negative, which was confirmed by fluorescence in situ hybridization. Normal karyotype was found in 32 B-ALL patients, hyperploidy was observed in 13 patients, hypoploidy was observed in two patients, complex karyotype was observed in five patients, t(1;19)(q23;p13) was observed in three patients, and other karyotype abnormalities were observed in 20 patients. KMT2A rearrangements were found in five patients. In the RALL-2016 protocol, only mutations of the KMT2A gene were considered as an adverse risk factor. Therapy was not changed depending on MRD status.

### 2.2. MRD Detection

MRD was carried out at the National Medical Research Center for Hematology (Moscow). Samples were sent from hospitals in Moscow, St. Petersburg, Nizhny Novgorod, Volgograd, Yaroslavl, Kaluga, Kirov, Surgut, and Yekaterinburg. Delivery of samples was carried out within 24 h from the moment of taking the bone marrow specimens. Bone marrow samples were collected in EDTA-treated tubes, lysed with Pharm Lyse solution (BD Biosciences), and centrifuged at 400× *g* for 3.5 min. The cell pellet was washed with 2 mL of CellWASH solution (BD Biosciences) and centrifuged. Monoclonal antibodies were added to 100 μL of cell suspension containing at least 2 × 10^6^ cells, if possible, according to Appendix A. After incubation, cells were washed and analyzed on a six-color FACSCanto II (BD Biosciences) or 13-color CytoFLEX (Beckman Coulter) flow cytometer. Detection of MRD was performed using the DfN approach previously described [22,24], as the centralized initial immunophenotypic study of blast cells was not included in the RALL-2016 study. Laboratories in local hospitals independently performed immunophenotyping for line verification of acute leukemia. However, LAIP was detected before treatment in patients who were treated at the National Medical Research Center for Hematology, and this LAIP was a starting point in the search for MRD.

MRD was defined as a population that consisted of 20 or more cells with an aberrant immunophenotype. The sensitivity of the method was at least 0.01%.

### 2.3. MRD Immunophenotype Description

Since MRD was detected on different flow cytometers using three different monoclonal antibody panels, a strategy for qualitative immunophenotype description was adopted from a previously reported study [25]. The expression of antigens was assessed in comparison with normal hematopoietic cells. Six variants of antigen expression were identified: (i) absence of antigen (−); (ii) partly positive (−/+); (iii) low expression (dim); (iv) partly negative (+/−); (v) positive (+); (vi) high expression (high). Partly positive expression was called if the proportion of positive cells was 50% or less and partially negative if more than 50%. The expression of CD10, CD34, CD38, CD20, and CD58 in MRD cells was assessed in comparison with normal B-cell progenitors, mature B cells, and plasma cells (PCs) (Figure 1). High expression of CD10 and CD34 in MRD cells was considered if it was higher than in normal B-cell precursors (hematogones). Dim expression of CD10 and CD34 in MRD cells was considered if it was homogeneous and lower than in hematogones. High expression of CD58 in MRD cells was considered if it was higher than it was in plasma cells (CD19^+^ cells with the strongest CD38 expression). Negative expression of CD38 in MRD was considered if it was as on mature B cells and did not overlap with hematogones. If MRD cells expressed CD38 higher than mature B cells, but lower than hematogones, this was denoted as “dim” CD38 expression. In this case, MRD and hematogones overlapped partly in CD38 expression. Positive expression of CD38 in MRD was considered if it was as in hematogones or higher.

The expression of CD19 and CD45 in MRD cells was assessed in relation to granulocytes, monocytes, and lymphocyte–leukocyte subsets presented in each sample (Figure 2). High CD19 expression was considered if it was higher than in hematogones and even in mature B cells. If MRD cells and mature B cells expressed CD19, the CD19 expression was labeled as “positive”. CD19 expression was considered as dim if it was lower in MRD cells than in mature B cells. Negative expression of CD19 in MRD cells was considered if it did not differ from negative control cells (CD19-negative lymphocytes, monocytes, and granulocytes). There was only one case of CD19-negative B-ALL in our study (initial diagnosis of B-ALL was confirmed by strong expression of CD10, CD22, and cytoplasmic CD79a). CD45 expression was considered negative in MRD cells if it was lower than on granulocytes and the MRD population did not overlap with the left edge of granulocytes in the SSC vs. CD45 dot plot. CD45 expression in MRD cells was considered positive in MRD cells if the blast cell population overlapped with lymphocytes on the SSC vs. CD45 dot plot. Intermediate variants between negative and positive CD45 expressions were “−/+” or “dim”, depending on the size of the negative part of CD45 expression.

### 2.4. Statistical Analyses

Statistical analyses were performed using IBM SPSS v.23 and R 3.6.3 software. Analysis of relapse-free survival (RFS) was performed using the Kaplan–Meier method. The initial date was the date of the MRD assessment. The time interval was counted from the initial date to the date of the first adverse event (relapse/death) or the date of the last contact (censoring). Cluster analysis of cases according to immunophenotype MRD was performed using the ward.D method of the dendextend package in R 3.6.3. To do this, a qualitative description of MRD immunophenotype was converted into a numerical semiquantitative equivalent: (i) (−) → 1; (ii) (−/+) → 2; (iii) (dim) → 3; (iv) (+/−) → 4; (v) (+) → 5; (vi) (high) → 6. The comparison of MRD values between two groups was performed using the Mann–Whitney test, while the comparison among three groups was performed using the Kruskal–Wallis test. Nonparametric tests were used due to the non-normal distribution of the data. The normality of distribution was tested using the Shapiro–Wilk test.

## 3. Results

The proportion of MRD-positive cases was 39.4%, 18.0%, and 13.2% in B-ALL patients on days 70, 133, and 190 respectively. The 4 year RFS was lower in MRD-positive patients on day 70 (37% vs. 78%, *p* < 0.0001) and on day 190 (19% vs. 78%, *p* = 0.0085) (Figure 3).

Analysis of the immunophenotype was performed only for MRD detected on day 70. MRD cases formed three distinct clusters on dendrogram. The first (n = 13) and second (n = 8) clusters were characterized by low expression of CD10 (“−” or “−/+”), while the third (n = 16) cluster consisted of CD10-positive cases (“+/−”, “+”, “high”). All cases in the second cluster were CD38-negative, whereas, in the first cluster, seven cases were CD38^+^, four were CD38^dim^, and only two cases were CD38^−^. High expression of CD58 was observed in the first cluster more frequently than in second one (five vs. zero cases) [Figure 4a].

RFS of patients depended on the belonging of MRD (on day 70) to the clusters of the obtained dendrogram [Figure 4b]. In patients with MRD of the first cluster, the median of RFS was 8.7 months; in patients with MRD of the second cluster, it was 37.1 months; in patients with MRD of the third cluster, it was 34.7 months (*p* = 0.04). MRD amount did not differ significantly across the three clusters. In the first cluster, the median MRD percentage was 0.13%; in the second, it was 0.06%; in the third, it was 0.10% (*p* = 0.97).

In the first cluster, three patients had KMT2A rearrangements, one had hypoploidy, and five had other karyotype abnormalities (der(21), del(9), del(11), der(7;9), and t(11;14)). The second cluster included two cases with complex karyotype and one case with t(2;14). In the third cluster, two patients had hyperploidy, two patients had complex karyotype abnormalities, and add(6), der(19), der(7) were found in three patients.

Because the clusters differed mainly in the expression of CD10, CD38, and CD58, the effect of the expression of these antigens on RFS was separately tested (Figure 5).

In patients with CD10^−^ MRD, the 4-RFS was lower than in patients with MRD with any positive part of CD10 expression (19% vs. 51%, *p* = 0.004). MRD percentages did not differ in these two groups (medians 0.11% vs. 0.07%, *p* = 0.67). Despite a CD10-negative immunophenotype of MRD, seven patients had B-II (common) ALL, and seven other patients had B-I (pro-B) ALL.

Patients with CD38-positive MRD had a lower 4-RFS compared to CD38-negative cases (0% vs. 51%, *p* < 0.0001). MRD amount did not differ in CD38-positive and CD38-low groups (medians 0.17% vs. 0.07%, *p* = 0.89).

High expression of CD58 in MRD cells also worsened the 4-RFS (21% vs. 40%, *p* = 0.02). MRD percentages did not differ in cases with or without high expression of CD58 (medians 0.03% vs. 0.16%, *p* = 0.51).

We assessed the influence of CD10, CD38, and CD58 expression combinations on RFS. Patients who had CD10^−^CD38^+^CD58^high^ MRD (the most adverse group) had RFS 0%. The hazard ratio was 15.1 (3.7–62.3). The 4-RFS was lower in patients who had at least one of CD10^−^, CD38^+^, or CD58^high^ (28% vs. 51%, *p* = 0.014; Figure 6). The hazard ratio was 4.3 (1.2–14.9).

## 4. Discussion

MRD detected by MFC is a strong prognostic marker in ALL, as demonstrated in the presented study and in many others [6,7,8,9,12,26]. Cytostatic intensity and treatment regimen differed between different clinical trials, but comparable MRD negativity between the RALL-2016 study and previously reported data was observed. The proportions of MRD-negative B-ALL patients included in the GMALL (German Multicenter Study Group for Adult Acute Lymphoblastic Leukemia) protocol were 66% and 78% at the end of induction and consolidation, respectively [10]; in the presented study, the proportions of MRD-negative patients were 60.6% and 82.0% at the same time periods. At the end of phase 2 induction, the proportion of MRD-negative non-T-ALL patients was 68.8% in the UKALL XII/ECOG2993 study (United Kingdom arm of the international ALL trial XII/Eastern Cooperative Oncology Group) [16]. At the end of consolidation, the proportion of MRD-negative patients in the presented study was also comparable to the PETHEMA (The Programa Español de Tratamientos en Hematología) ALL-AR-03 trial, there it was 86% [9].

The 4 year RFS was lower in MRD-positive patients on days 70 and 190; however, RFS did not differ significantly depending on MRD measured on day 133. This fact could be explained by a small number of patients who were MRD-positive on days 133 and 190 of the protocol. There was a statistical trend (*p* = 0.064) in differences in 4 year RFS between patients depending on MRD on day 133. It is likely that, with a larger number of patients enrolled to trial or a longer follow-up period, these differences will become more pronounced.

MFC is able not only to enumerate MRD accurately, but also to describe an antigen expression profile of residual blast cells. We showed that CD10^−^, CD38^+^, and CD58^high^ MRD is associated with worse prognosis in B-ALL patients. No studies on the MRD immunophenotype have been published thus far; however, there are a number of studies devoted to the study of the prognostic significance of the initial immunophenotype of blast cells before treatment [27,28,29,30].

CD10 is an antigen expressed by B-cell precursors and a subset of mature neutrophils [31,32]. In B-ALL, CD10 expression allows identifying B-ALL subtypes [23]. CD10-negative BI (pro-B) ALL is associated with KMT2A rearrangement, which is an adverse prognostic marker [33,34]. According to a previous study, CD10-negative patients with BIII (pro-B) ALL had better a prognosis than CD10-positive ones [35]. In the presented study, patients with CD10^−^ MRD had lower RFS; however, only half of them had B-I (pro-B) ALL, and only three patients had KMT2A rearrangements. The other half of these patients had BII (common) B-ALL. We assume that, during therapy, the CD10-negative population of blast cells is selected, which is likely to be more malignant than the CD10-positive population.

One study showed that CD38 positivity of blast cells correlated with the presence of ETV6::RUNX1 rearrangement, and a higher number of B-ALL cells with low CD38 expression could be an early indicator of relapse risk [29,30]. CD38 is a multifunctional transmembrane glycoprotein that operates both as a receptor and as an enzyme. CD38 is involved in the metabolism of nicotinamide dinucleotide and is also a regulator of intracellular calcium homeostasis [36]. Plasma cells have the highest expression of CD38, but B cells, T cells, NK cells, myeloid cells, and precursors also express CD38 with different density [37]. CD38 is involved in B-cell differentiation and proliferation, and the CD38 expression is positive in B-cell precursors. However, B-ALL cells have a reduced expression of CD38 in most cases [38]. In the presented study, in 26 (70%) cases of MRD detected at the end of induction, CD38 expression was dim or negative. However, positive CD38 expression in MRD cells was associated with worse disease-free survival (DFS). We did not find studies on the effect of CD38 expression on the prognosis of B-ALL patients, but the significance of CD38 expression has been well studied in lymphoproliferative diseases. For example, in 27–46% of patients with chronic lymphocytic leukemia (CLL), CD38 expression was found on tumor B cells [39,40]. The presence of CD38 in CLL is associated with a worse survival prognosis, as well as an increase of CD38 expression on tumor cells, which reflects a switch to a more aggressive CLL type [41].

CD58 is an intercellular adhesion molecule that binds to the CD2 antigen which is expressed on cells. CD58 expression is found on many hematopoietic and nonhematopoietic cells, as well as malignant neoplasms, including CLL, Hodgkin’s lymphoma, multiple myeloma, and acute myeloid leukemias [42]. CD58 is often expressed higher on leukemia B lymphoblasts than on normal B-cell progenitors [38,43,44]. It is assumed that a high density of CD58 expression leads to an increase in the adhesive properties of blast cells, their proliferative activity, and the tendency to undergo apoptosis. High proliferative activity of MRD cells with high CD58 expression could explain the deterioration of DFS found in the presented study. However, another study found that CD58 negativity of initial blast cells was a predictor for adverse outcome in B-ALL patients [30]. Interestingly, in patients with pancreatic ductal adenocarcinoma, CD58 was upregulated in cancer tissues and associated with worse overall survival and disease-free survival [45].

The obtained data should be confirmed in subsequent studies, given the limitations of the present investigation, such as the low number of MRD-positive cases. Expanding the spectrum of antigens under study and comparing the immunophenotype with more detailed cytogenetics and molecular features of blast cells seems to be interesting perspective topic of research.

It should be noticed that antigen expression changes occur during treatment. For example, downmodulation of CD10 and CD34 and upmodulation of CD20 and CD45 during induction therapy was confirmed in different studies [46,47]. Thus, the prognostic significance of MRD immunophenotype and initial blast cell immunophenotype could be different.

## 5. Conclusions

In summary, the present study revealed that MRD cells detected by MFC at the end of induction could be clustered according to their immunophenotype, with CD10, CD38, and CD58 expression making the greatest contribution to clustering. The absence of CD10, the presence of CD38, and the high expression of CD58 by MRD cells were associated with worse DFS. These findings indicate that MRD immunophenotype could be an additional prognostic factor in B-ALL patients.

## Figures and Tables

**Figure 1 diagnostics-13-00021-f001:**
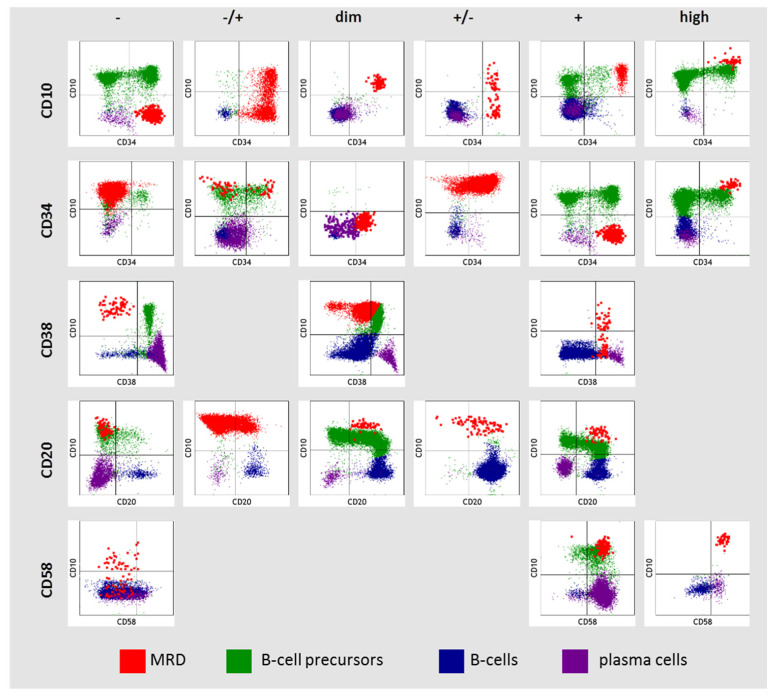
Description of CD10, CD34, CD38, CD20, and CD58 expression in residual blast cells (MRD—measurable residual disease).

**Figure 2 diagnostics-13-00021-f002:**
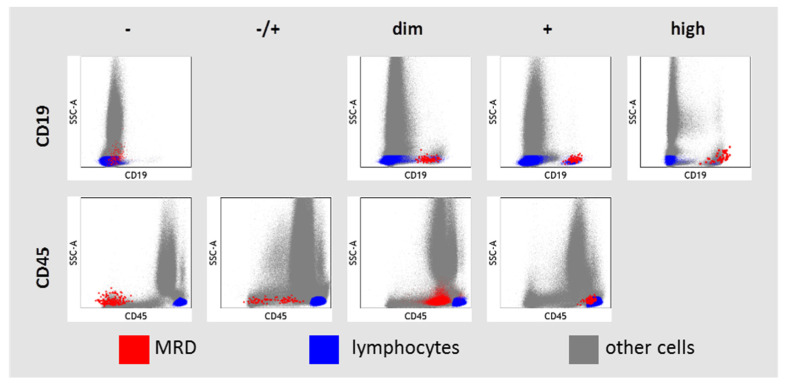
Description of CD19 and CD45 expression in residual blast cells (MRD—measurable residual disease).

**Figure 3 diagnostics-13-00021-f003:**
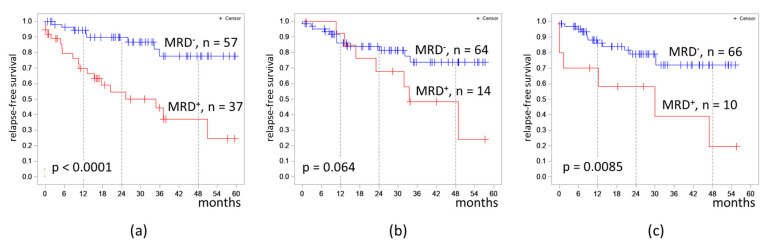
Relapse-free survival of patients with B-cell acute lymphoblastic leukemia depending on MRD at different timepoints: (**a**) at the end of induction II (day 70); (**b**) at the end of consolidation III (day 133); (**c**) at the end of consolidation V (day 190). The initial date was the date of the MRD assessment.

**Figure 4 diagnostics-13-00021-f004:**
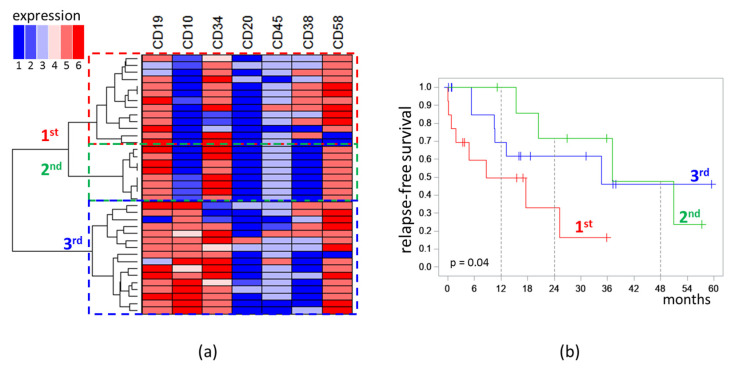
Analysis of immunophenotype of measurable residual blast cells detected at the end of induction: (**a**) Dendrogram and heat map of semiquantitative immunophenotype description (1: “−”; 2: “−/+”; 3: “dim”; 4: “+/−”; 5: “+”; 6: “high”); (**b**) relapse-free survival depending on belonging to the clusters highlighted on the dendrogram. The initial date was the end of induction.

**Figure 5 diagnostics-13-00021-f005:**
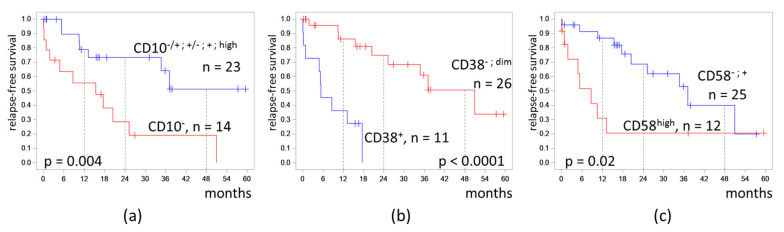
Relapse-free survival of patients depending on expression of CD10 (**a**), CD38 (**b**), and CD58 (**c**) on measurable residual blast cells detected at the end of induction. The initial date was the end of induction.

**Figure 6 diagnostics-13-00021-f006:**
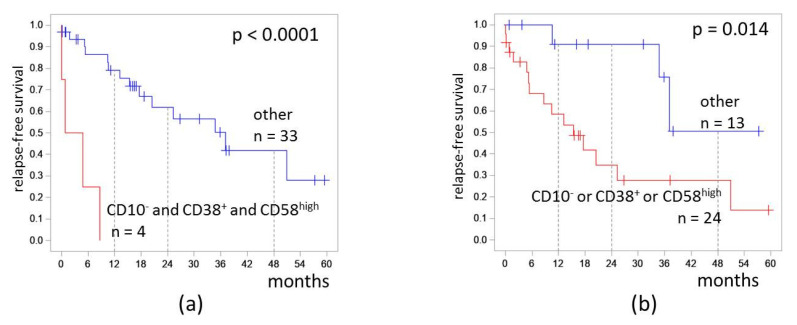
Relapse-free survival of patients depending on expressions of CD10, CD38, and CD58 on measurable residual blast cells detected at the end of induction: (**a**) combination of CD10^−^ and CD38^+^ and CD58^high^; (**b**) CD10^−^, CD38^+^, or CD58^high^ immunophenotype. The initial date was the end of induction.

## Data Availability

Data are not shared due to privacy restrictions.

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
