# Peer review of "Immunophenotype of Measurable Residual Blast Cells as an Additional Prognostic Factor in Adults with B-Cell Acute Lymphoblastic Leukemia"

_diagnostics, 2022, doi:10.3390/diagnostics13010021_

Round 1

Reviewer 1 Report

As the authors describe, MFC-MRD is faster, cheaper and applicable to most ALL cases in comparison with PCR. Two different MFC approaches are used; the first is the determination of leukemia-associated immunophenotype of blasts cells before treatment and, consequently, in follow-up samples. The second, DfN, requires strong knowledge of antigen expression patterns of normal hematologic progenitors, and does not require a pretreatment sample. The difference is that MFC is less standardized compared with PCR. So, the lack of uniformity for FCM-MRD, between laboratories, could be an important limitation; MFC-MRD should be defined by laboratories with a specific and robust expertise. Currently, in the routine clinical care, it should be important to apply rigorous analysis which can ensure interlaboratory reproducibility. However, the relation between MRD-immunophenotype and patients’ survival had not been studied yet. In this paper the authors studied a large number of patients with ALL; the MRD assessment, performed in patients in CR remission, did not cause deviations from the therapeutic protocol.

The MRD immunophenotype description is complete and well reported. The results achieved are comparable to those reported in previously published papers. These results should be confirmed in subsequent studies, and immunophenotype should be compared with cytogenetic and molecular studies. The authors, also describe the leukemic cells antigen expression changes during treatment and comment that MRD cells detected by MFC at the end of induction, could be clustered according to their immunophenotype with CD10, CD38, and CD58 expression making the greatest contribution to clustering. The absence of CD10, the presence of CD38, and the high expression of CD58 by MRD cells are associated with worse DFS. These findings indicate that MRD-immunophenotype could be an additional prognostic factor in B-ALL patients.

The results of the different MRD techniques must be published and shared and integrated with clinical data with the objective of improving quality of treatment and consequently patients’ outcome.

Author Response

We would like to thank the reviewer for taking the necessary time and effort to review the manuscript.

Reviewer 2 Report

Dear Authors,

Congratulations for your work. Here are my comments:

Please remove numbers from abstract and present it as dictated by journal guidelines. Please use brackets’ syntax correctly i.e. not ((X)) but [(X)] and not (X)(Y) but (X;Y).

Line 15: According to https://clinicaltrials.gov/ct2/show/study/NCT03462095 this clinical trial should have come to an end by now, recruiting 350 participants. Please explain in text reasons for slow accrual. Of interest, principal investigators Valeriy Savchenko and Olga Gavrilina in this trial are not among the authors of this publications. Why so?

Line 16: “6-“ instead of “6”

Line 17: why have you chosen 4-year rates instead of 5- or 10-year measures that can be easily reproduced and compared?

Lines 18-19: Definitions are different in text. Please rephrase: high CD58 is OK, but “high expression of CD38” is not accurate

Line 34: please omit “the”

Line 52: please change “an” to “the”

Line 63: B-ALL cases account for 52.6% of the cohort. Does this discrepancy appear in your initial cohort? Or is it because of the CR-guided selection?

Line 67: MRD was conducted in 94 B-ALL patients at the end of induction II, but you sample size was 102. These 8 patients were deceased or what until induction II?

Line 99: “Six” instead of “6” in the beginning of the sentence

Lines 104, 110, 111: “B-cells” / “B cells” are not used uniformly throughout the text

How do you explain the fact that there was no significant difference in 4-year RFS between MRD-positive and MRD-negative patients at the end of consolidation III (while it was evident for D70 and D190)?

In Figure 5 refers to 37 MRD cases, which implies that you used non-parametric tests. Is it so? Were the MRD values normally distributed?

In support of your conclusion, it would be of great interest for the readership to combine your findings in one plot and analysis (CD10-, CD38+, and high CD58) and calculate either positive predictive value or diagnostic odd ratios. By all means, it would be confirmatory of your results to describe these findings by combining them.

Author Response

We thank the reviewer for thoughtful review of our work and kind words.

Round 2

Reviewer 2 Report

Dear authors,

thank you for incorporating my suggestions in your text. I wish you are now more content about your manuscript.